# Filter Papers to Collect Blood Samples from Dogs: An Easier Way to Monitor the Mass Vaccination Campaigns against Rabies?

**DOI:** 10.3390/v14040711

**Published:** 2022-03-29

**Authors:** Marine Wasniewski, Jacques Barrat, Samia Ben Maiez, Habib Kharmachi, Mariem Handous, Florence Cliquet

**Affiliations:** 1ANSES, Nancy Laboratory for Rabies and Wildlife, European Union Reference Laboratory for Rabies Serology, European Union Reference Laboratory for Rabies, WHO Collaborating Centre for Research and Management in Zoonoses Control, OIE Reference Laboratory for Rabies, Technopôle Agricole et Vétérinaire, CS 40009 Malzeville, France; jlbperso690@gmail.com (J.B.); florence.cliquet@anses.fr (F.C.); 2Laboratory of Rabies, Pasteur Institute of Tunis, 13 Place Pasteur, Tunis 1002, Tunisia; samia.benmaiez@pasteur.tn (S.B.M.); h.kharmachi@yahoo.fr (H.K.); mariem.handous@pasteur.tn (M.H.)

**Keywords:** rabies virus, filter paper, dogs, ELISA, FAVN test, mass vaccination

## Abstract

Rabies is a deadly viral disease present mainly in low-income countries of Africa and Asia. Dogs are the main reservoir and the source of human deaths. Mass vaccination campaigns of dogs are pivotal to achieve rabies elimination. The monitoring of the immune response of the dog population is necessary to evaluate the effectiveness of these campaigns, taking into account field conditions. This study explores the feasibility and the performance of a new tool using filter papers (FPs) to collect blood samples associated with an Enzyme-Linked ImmunoSorbent Assay (ELISA) titration of rabies antibodies in dogs. A total of 216 eluates from FP samples were collected from 111 dogs kept in experimental facilities in France and 29 dogs from the field in Tunisia. Sera were also analyzed using both the Fluorescence Antibody Virus Neutralization test (FAVNt) and ELISA. A high specificity (98.0%) was obtained by testing FP blood eluates from 51 unvaccinated dogs, with the results compared with those of FAVNt and ELISA on serum samples. The coefficients of concordance between FP eluates and tested sera were 88.9% for FAVNt and 88.0% for ELISA. Blood filter papers coupled with the titration of rabies antibodies by ELISA provide a reliable, simple, and effective solution to overcome the issues of the logistics and transport of samples, especially in low-income countries.

## 1. Introduction

Rabies is present on all continents except Antarctica and some areas that have remained historically free of the disease due to their favorable geographical location, such as certain islands in the Pacific Ocean [1]. Canine rabies is caused by the rabies virus (RABV), the prototype virus of the Lyssavirus genus [2,3]. Domestic dogs serve as a major reservoir for RABV in many low-income countries and are able to maintain the transmission of the virus in a well-defined cycle [4]. They are referred to as a dynamic reservoir because the natural maintenance cycle of the virus in endemic and enzootic areas is from dog to dog, while individually infected dogs die [5]. Dogs are responsible of the vast majority of human cases (98–99%) [5,6,7], mainly in low-income countries located in Asia and Africa [8,9].

In this context, animal rabies control is considered as one of the most effective means of preventing rabies in humans. Thus, to reduce the risk of human exposure to the rabies virus, it is essential to decrease the virus circulation in domestic animals. In recent years, disease elimination programs using the mass vaccination of dogs have greatly reduced or even eliminated human rabies cases in several countries [7,10,11]. For example, South America has made tremendous progress in eliminating rabies and has achieved a reduction of over 90% in human deaths and dog cases since 1983 [12]. This is the result of the implementation of rabies elimination programs combining the mass parenteral vaccination of dog populations with the systematic notification of human cases [13].

Passive surveillance should be associated with the monitoring of the seroconversion of vaccinated animals to check and assess the performance of mass vaccination campaigns from an immunological point of view [7]. Several methods could be used to assess the seroconversion in vaccinated animals, such as the Enzyme-Linked ImmunoSorbent Assay (ELISA) or the seroneutralization methods, e.g., the Fluorescent Antibody Virus Neutralizing test (FAVNt) [14] and the Rapid Fluorescent Focus Inhibition Test (RFFIT) [15]. As a faster and easier method to implement, ELISA is preferred for vaccination follow-up in countries having disease elimination programs such as the mass parenteral vaccination of dogs and the oral vaccination of wildlife [16,17,18,19,20].

Even if the serological test is appropriate, several problems can occur during the sampling of the dogs in the field, as well as for wildlife samples, making the analyses difficult or even not feasible. In some areas, the blood samples are not processed on the spot, as no equipment is available, and there is a need to send or ship the blood vials to a laboratory for their treatment prior to analysis. This could clearly increase the risk of broken or not properly closed tubes in the end. Moreover, there is a risk of injury for people, as they use syringes and needles for blood collection by venipuncture on conscious animals that are quite aggressive.

Thus, improving the sampling method would help improve the monitoring of mass vaccination campaigns. Several years ago, we developed a method for collecting blood samples in wildlife using filter papers, which simplifies sample collection, transport, and storage [21]. The satisfactory results obtained in wildlife for this blood-sampling method encouraged us to transpose it to monitor the rabies mass vaccination campaigns in dogs. The objective of this preliminary study was to assess the feasibility of this method in dogs, as well as its reliability and consistency, before investigating its use at a large scale.

## 2. Materials and Methods

### 2.1. Samples

#### 2.1.1. Dog Samples and Ethical Aspect

##### In France

Blood samples from 111 dogs were collected in the Anses-Nancy experimental station (Atton, Meurthe-et-Moselle department, France). Fifty-one blood samples were collected from unvaccinated dogs and eighty-two samples were obtained from vaccinated dogs. According to some experimental protocols, 22 vaccinated dogs were sampled twice, 7 days apart.

In the experimental station, all dogs were housed in similar conditions and were already involved in research projects. Experiments and husbandry were conducted in compliance with European Directive 2010/63/EU [22] and French regulations on ethics in animal experimentation. These experiments were approved by the Anses/ENVA/UPEC ethics committee and the French Ministry of Research (Apafis n° 3164-2015121414072410).

##### In Tunisia

Blood samples from 29 dogs from the field (except one dog maintained under experimental conditions) were collected at 4 different times before and after the 2019 rabies mass vaccination campaign (D-21, D0 (day of vaccination), D15, and D30). Filter papers and serum samples were both available for 83 samples. A total of 79 samples obtained during the rabies mass vaccination campaign were analysed, whereas 4 samples collected 15 days apart came from the same animal kept under experimental conditions.

Animal experiments were conducted with the approval of the Biomedical Ethics Committee of the Pasteur Institute of Tunis (2017/04/I/LR11IPT03/V1). In all of the vaccination sites, informed consent was obtained prior to each survey as well as a blood test from the dog owners, who were fully informed of the purpose, approach, and progress of the study. For all dogs, vaccination and blood sampling were carried out with the presence of their owners and their express consent.

#### 2.1.2. Blood Sample Collection

##### In France

The same Filter Paper (FP) as the one used for the wildlife study (Bio-Rad Mini Trans-Blot Filter Paper, catalogue ref. 170-3932, Hercules, CA, USA) was selected for the collection of dog blood samples [21]. The same surface of the FP (10 cm × 7.5 cm) was saturated with whole blood, as described below.

Blood samples were taken from the jugular or cephalic vein using a syringe and a needle on conscious animals and collected into two untreated tubes. One tube was used to saturate the FP, whereas the second tube was centrifuged to obtain the serum sample. Serum samples were heat-inactivated (56 °C, 60 min) and stored at −20 °C before being tested. The heat-inactivation step was required as these samples were tested by the seroneutralization test.

##### In Tunisia

Blood samples were collected from the radial vein using a syringe and a needle on conscious animals gently held by the owner. For each dog, about 5 mL of blood was placed in a plastic serum tube and about 7.5 mL of blood was used to impregnate the FP.

Samples were transported from the field to the laboratory in a cooler. Then, tubes were centrifuged to obtain the serum and stored at −20 °C.

Frozen serum samples were sent to Anses-Nancy laboratory, heat-inactivated (56 °C, 60 min), and stored at −20 °C before being tested.

#### 2.1.3. Processing of the FP

##### In France

After the impregnation step, FPs were air-dried at room temperature for at least 4 h, then individually stored at room temperature in a paper envelope labelled with the animal number and date of collection.

##### In Tunisia

After the impregnation step, FPs were air-dried at room temperature on aluminum foil for at least 4 h. To facilitate the drying, they were returned to foil after 2 h. Then, the FPs were stored in the same conditions as described above. FPs were sent to Anses-Nancy laboratory.

#### 2.1.4. Obtaining Eluates from FPs

As for wildlife blood samples, a surface of 7.5 cm^2^ was considered sufficient for serological testing [21]. The processing of the FP strip to perform a test was carried out as previously described by Wasniewski et al. [21]. Briefly, a 1 cm × 7.5 cm strip of FP was cut with clean scissors and then placed into a tube containing 890 µL of Dulbecco’s modified eagle medium (DMEM) supplemented with antibiotics for the elution step. The strip was pressed against the walls of the tube to allow full contact with the elution medium. The strip was incubated in the medium for around 1 h 30 min. After centrifugation (800× *g* for 30 min), the supernatant (eluate) was collected and stored at −20 °C until analysis [21].

This processing was performed to treat all FPs in Anses-Nancy laboratory.

The eluates obtained from FPs were tested only by ELISA due to the cytotoxicity of the FP eluates on the cell layer, while serum samples were tested by both FAVNt and ELISA.

As for the wildlife study, the predilution of the FP eluates was not considered for the calculation of titers when performing ELISA testing [21].

### 2.2. Methods Used

#### 2.2.1. FAVN Test

Rabies-neutralizing antibodies were determined by using the FAVNt, as described by Cliquet et al. [14]. Reagents used are described in Cliquet et al. [23]. The neutralizing titers are expressed in international units per milliliter (IU/mL) by comparing results obtained with the serum sample to those of the positive reference serum. The threshold of positivity was 0.5 IU/mL.

#### 2.2.2. ELISA Test

The BioPro Rabies ELISA Ab kits were obtained from the BioPro company (Prague, Czech Republic). Serum samples were titrated according to the manufacturer’s instructions as previously described by Wasniewski et al. [24].

The conditions of validation provided by the manufacturer were strictly followed for the interpretation of the results for the different samples, as previously described by Wasniewski et al. [24]. The percentage of blocking (%PB) was calculated according to the manufacturer’s instructions (i.e., %PB = ([ODNC – Odsample]/[ODNC – ODPC]) × 100), where ODNC is the optical density of the negative control, ODPC is the optical density of the positive control, and ODsample is the optical density of the tested sample.

In the context of international trade, for this ELISA kit, a positive titer for rabies antibodies is indicated by a signal equal to or greater than 70%PB, and a negative titer is indicated by a signal of less than 70%PB, as recommended in the manufacturer’s specifications. According to the manufacturer, this 70%PB corresponds to the 0.5 IU/mL threshold. For monitoring the effectiveness of oral vaccination campaigns in wildlife, the manufacturer has reduced the threshold of positivity to 40% to be able to detect a seroconversion level.

### 2.3. Statistical Methods

#### 2.3.1. Specificity

Specificity was expressed as the proportion of true negative results by titrating FP eluates and serum samples of the 51 dogs caged in the experimental station (i.e., <0.5 IU/mL for the FAVNt and <70% for the BioPro ELISA).

#### 2.3.2. Coefficient of Concordance

The agreements of the different methods used in this study were evaluated by pairing the results obtained from FP eluates by using the ELISA test with those obtained from the serum samples by using the ELISA test and the FAVNt. The ratio of samples determined to be positive or negative with both methods divided by the total number of tested samples represented the coefficient of concordance.

#### 2.3.3. Calculation of the 95% Confidence Interval (CI) for a Proportion

The 95% CI was calculated online using the Binomial “exact” calculation [25].

## 3. Results

The results obtained by the FAVNt were considered as reference results.

### 3.1. Specificity

The percentage of specificity for the FP method was equal to 98.0% when compared to the FAVNt and ELISA test performed on serum samples from unvaccinated dogs. Indeed, among the 51 negative samples, one eluate was found positive by the ELISA test (%PB = 88.8). The corresponding serum was also found positive by ELISA test, with the same percentage value. Moreover, another serum sample was found positive by ELISA (%PB = 94.6), while the eluate was found negative (%PB = 16.5). These two samples given discrepant results were collected in the experimental station from caged unvaccinated dogs belonging to the same experiment.

### 3.2. Coefficient of Concordance

#### 3.2.1. In France

The coefficient of concordance between the FP method combined with the FAVNt on serum samples was equal to 94.0% (Table 1), while it was equal to 87.2% between the FP eluates and serum samples when using the ELISA test (Table 2).

##### FP Eluates Combined with ELISA versus FAVNt on Serum Samples

Eight samples had discordant results among the 133 tested samples (Table 1).

Three eluates were found positive (%PB = 79.6, 76.3, and 88.8), whereas the paired serum samples were found negative by using the FAVNt. Indeed, the first two corresponding serum samples had a titre of 0.38 IU/mL, which is very close to the threshold of positivity, and the third had a titre of 0.06 IU/mL. These serum samples were also found positive with the ELISA (%PB equal to 79.6, 89.6, and 88.8, respectively) (Table 2).

Five eluates were found negative (with %PB equal to 61.6, 57.3, 34.2, 56.0, and 54.3), whereas the corresponding serum samples had positive FAVNt titres, equal to 0.87, 0.50, 0.66, 0.50, and 1.15 IU/mL, respectively.

Among these five eluates, four corresponding sera were found positive by ELISA. For the five corresponding sera, the %PB were equal to 89.3, 84.3, 69.4, 84.9, and 85.2, respectively.

##### FP Eluates versus Serum Samples Tested Using ELISA

Seventeen samples had discordant results (Table 2). For all of them, FP eluates were found as negative (%PB ranging from 29.5 to 65.7), whereas the serum samples were found positive by ELISA (%PB ranging from 74.1 to 93.3).

Among these seventeen eluates, four corresponding sera were found positive by FAVNt (titres ranging from 0.5 to 1.15 IU/mL). The other sera (*n* = 13) were found negative by FAVNt (titres ranging from 0.22 to 0.38 IU/mL for 12 samples and one sample had a titre equal to 0.06 IU/mL (see the Specificity section)).

#### 3.2.2. In Tunisia

The coefficient of concordance between the FP method combined with the ELISA test and FAVNt on the paired serum samples was equal to 80.7% (Table 1), while it was equal to 89.2% between the FP eluates and serum samples when using the ELISA test (Table 2).

##### FP Eluates Combined with ELISA versus FAVNt on Serum Samples

Sixteen samples had discordant results among the 83 tested samples (Table 1).

Three eluates were detected as positive (%PB = 87.0, 70.2, and 79.9), whereas the corresponding sera were found as negative by using the FAVNt. These serum samples had titres equal to 0.29, 0.17, and 0.38 IU/mL, which are all very close to the threshold of positivity, and were found positive with the ELISA test (%PB equal to 92.0, 86.8, and 90.0, respectively).

##### FP Eluates versus Serum Samples Tested Using ELISA

Nine samples had discordant results. For all of them, the FP eluates were found as negative, with %PB between 29.9 and 67.1, whereas the serum samples were found as positive by the ELISA test, with %PB ranging from 70.3 to 89.3.

The four other corresponding sera were found negative by FAVNt, with titers ranging from 0.22 to 0.38 IU/mL.

#### 3.2.3. Overall

For the 216 samples collected from vaccinated and unvaccinated dogs in France (*n* = 133) and in Tunisia (*n* = 83), the coefficient of concordance between the FP method combined with the FAVNt on serum samples was equal to 88.9%, while it was equal to 88.0% between the FP eluates and serum samples when using the ELISA test (Table 3).

## 4. Discussion

Rabies is a viral zoonosis leading to a high disease burden in numerous developing countries [7]. Around 59,000 human deaths occur per year, mainly due to dog bites [6,26]. Dogs are considered as the main reservoir in urban and rural areas in low-income countries [7,9]. In 2018, United Against Rabies, a collaboration between four partners (the World Health Organization, the Food and Agriculture Organization of the United Nations, the World Organization for Animal Health, and the Global Alliance for Rabies Control (WHO, OIE, FAO, and GARC)) launched a Global Strategic Plan to end human rabies deaths from dog-mediated rabies by 2030 [27]. In this plan, mass dog vaccination is highly recommended to save human lives by stopping the transmission of rabies at its source. Global efforts to reduce the risk of human exposure to the virus are currently made through mass vaccination programs of dogs living in urban but also in rural areas to decrease the incidence of rabies [6,28,29].

The Global Strategic Plan also recommends to enhance the monitoring of dog vaccination coverage [27]. This monitoring is achieved through the registration and temporary or permanent identification of dogs, if possible, to differentiate unvaccinated dogs for follow-up vaccination [7]. Serological monitoring is also encouraged by international organizations [7,30] in certain situations, i.e., if animal rabies cases are still reported despite regular annual vaccination, or if changes have occurred in the vaccination program, such as the use of a new vaccine or a suspicion of a break in the cold chain. Such monitoring aims to determine the seroconversion rates of vaccinated animals at the peak of the antibody response (around 28 days after the primovaccination) [7].

Vaccination coverage and immune coverage are indeed two key elements to consider for the success of the control strategy. Regular mass vaccination campaigns are conducted with the objective of reaching at least 70% vaccination coverage [31,32,33] to have a significant effect on disease control. As for wildlife samples, the major difficulties met in the field involve the storage of the blood samples prior to their shipment to the laboratory and the conditions of shipment, as well as the handling of nonobedient dogs during blood sampling. In low-income countries and, more especially, in rural areas or in areas that are difficult to access or have high temperatures, the logistics and equipment to treat blood samples are often missing. This could lead to an unsatisfactory storage of blood samples from certain vaccinated areas, which could result in unsatisfactory or missing results and therefore could cause an underestimation of the immunity coverage in the dog population after the rabies mass vaccination campaign.

We report in this study a method that we have already developed for wildlife samples collected in the field [21], consisting in the use of FP eluates associated with an ELISA to assess the immunity of dogs against rabies. In wildlife, we applied a piece of filter paper to the wound of the animal to collect blood without a needle or syringe. The FP technology has already been extensively published for collecting human blood and blood from several domestic and wild animal species (for a review, see [21]) for the detection of diseases. To the best of our knowledge, a study such as this, which reports a method of blood collection in field dogs for further rabies antibody testing, has never been published before. The objective of this work was to evaluate the feasibility and the performance of the sample collection method for dogs associated with the commercial BioPro ELISA for antibody testing. This ELISA has been previously validated for serum samples from different animal species, including dogs, cats, foxes, raccoon dogs, wild boars, African wild dogs, spotted hyenas, lions, leopards, and banded mongooses [24,34,35,36,37]. The OIE and WHO indicate that ELISAs are now also recognized as acceptable tests for monitoring antibody response of vaccinated animals in the framework of rabies control [7,30]. The FAVNt remains the reference test for the purposes of measuring antibody responses to vaccination prior to international animal movement or trade. Therefore, to assess the performances of the blood FP method, the sera of dogs enrolled in the study were tested with the BioPro ELISA and FAVNt.

The results of this study showed a high specificity (98%) for the test involving FP blood eluates obtained from 51 unvaccinated dogs, with the results compared with those of the FAVNt and ELISA on serum samples. One eluate was found positive by ELISA, while the serum sample of this dog gave discordant results when using both assays; it was found negative by FAVNt and positive by ELISA. Another serum sample was also found positive by ELISA only. This result is in accordance with a previous validation study of the BioPro ELISA on serum samples from dogs, determining a specificity of 100% [24]. Considering that the objective of this sampling method is to facilitate the collection and transport of biological material to further determine the level of antibodies for assessing the herd immunity of field dogs following vaccination, we consider that the specificity found in this study is satisfactory. Further assessment of the FP blood method on field dogs never vaccinated against rabies will be necessary to confirm this result obtained on caged experimental dogs.

The FP blood eluates and corresponding serum samples from unvaccinated and vaccinated dogs kept in experimental conditions showed percentages of concordance of 94% and 87.2% for sera tested using the FAVNt and ELISA, respectively. Interestingly, these percentages of agreement are close to those obtained in the study performed in foxes and raccoon dogs using the same method of sampling (i.e., between 91% and 95%) [21]. It should be noticed that this study was also performed on caged experimental foxes and raccoon dogs. On the contrary, regarding the FP eluates from vaccinated dogs in Tunisia, a high percentage of agreement was observed between FP eluates and serum samples tested with ELISA (89.2%), whereas this percentage was 80.7% between FP eluates and serum samples tested by the FAVNt. Experimental conditions are often considered as optimal when performing a study using seroneutralization assays, as compared to field conditions. This could explain the difference observed in the percentages of agreement and the best performances obtained under experimental conditions when using the FAVNt.

The overall agreement between all 216 tested FP eluates and the corresponding tested sera was 88.9% for FAVNt and 88.0% for ELISA. The data clearly show that the discordant results mainly occurred with the FP eluates that tested negative and the corresponding serum samples that tested positive, either with the FAVNt or ELISA. It should be noted that the agreement between the two serological tests was 87.2% for the 133 dog serum samples from the experimental station and 81.9% for the 83 field dog serum samples collected in Tunisia (Appendix A). In the study of Wasniewski et al. [24], this agreement between both tests measured on field cat and dog samples was 86.2%, corresponding to the results obtained on experimental dogs in this study. The lower agreement for the field dogs resulted in the fact that 15 samples among the 83 had discrepant results. Seven sera were found negative by FAVNt and positive by ELISA, and, conversely, 8 samples were found positive by FAVNt and negative by ELISA. It should be highlighted that among the seven false-positive results with ELISA, four samples had titres equal to 0.38 IU/mL; two had titres equal to 0.29 IU/mL, which is very close to the threshold of positivity; and the last one had a titre equal to 0.17 IU/mL. Moreover, for the eight false-negative results, the FAVNt titres ranged from 0.66 to 2.62 IU/mL. Three samples had titres very close to the threshold of positivity, between 0.66 and 0.87 IU/mL. The ELISA uses a crude glycoprotein G as the coating antigen, and it may therefore detect antibodies that are not responsible for rabies virus neutralization [17,24].

Although the data demonstrate a high specificity, with only one false-positive compared to the FAVNt, this study suggests a slight lack of sensitivity of the FP method as compared with other investigated serological tests, i.e., there is a risk of falsely determining that a dog is insufficiently protected. Therefore, the immunization coverage could be slightly under-estimated. For the objective of the post-vaccination monitoring to assess the seroconversion rate of vaccinated animals, the most important criterion is the specificity to avoid any false security.

As this study was the first to be conducted in the field on dogs, we conducted a protocol by impregnating the entire surface of the FP for each dog sampled to obtain a high number of strips for developing and validating the test. Therefore, the volume of blood required was high, as under field conditions in Tunisia it was found that the quantity of blood necessary to properly impregnate the full FP was at least 5 mL. The impregnation could be difficult in cases of rain or wind, as were faced during this study. As soon as the protocol is fully evaluated, a FP strip will be sufficient, requiring a blood volume of around 1–1.5 mL for each sampled dog. Furthermore, it could be useful to place the impregnated FPs into resealable plastic zipper bags to protect them from rain and also to allow their transport to the laboratory. Despite these limitations, the FP sampling method presents several advantages, such as no special equipment required, no laboratory equipment needed to centrifuge and store the samples, and the easy and cheap shipping and transport of FP papers in a simple envelope.

In conclusion, this preliminary study demonstrates the feasibility of the FP sampling method to collect field blood samples from dogs used in association with the BioPro ELISA to evaluate the level of the herd immune response following antirabies vaccination. The specificity is highly satisfactory and the results of the comparison between FP eluates and serological testing suggest percentages of agreement around 88%. Additional studies will be conducted in the field using filter paper strips and plastic bags (or small boxes) to proceed to a rigorous validation of this method. Furthermore, it will be essential to assess the specificity of the FP sampling method with field unvaccinated dogs.

## Figures and Tables

**Table 1 viruses-14-00711-t001:** FP eluates titrated with the BioPro ELISA kit versus sera titrated with FAVN test.

Assay	Result	BioPro Rabies ELISA Ab Kit—FP Eluates							
		Dogs—Experimental Conditions (France)				Dogs—Field Conditions (Tunisia)			
		Positive (≥70%)	Negative (<70%)	Overall Agreement	95% CI	Positive (≥70%)	Negative (<70%)	Overall Agreement	95% CI
FAVN test sera	Positive (≥0.50 IU/mL)	59	5			34	13		
	Negative (<0.50 IU/mL)	3	66	94.0%	(88.5–97.4)	3	33	80.7%	(70.6–88.6)
	Total	62	71			37	46		

**Table 2 viruses-14-00711-t002:** FP eluates titrated using the BioPro ELISA kit versus sera titrated using the BioPro ELISA kit.

Assay	Result	BioPro Rabies ELISA Ab Kit—FP Eluates							
		Dogs—Experimental Conditions (France)				Dogs—Field Conditions (Tunisia)			
		Positive (≥70%)	Negative (<70%)	Overall Agreement	95% CI	Positive (≥70%)	Negative (<70%)	Overall Agreement	95% CI
BioPro Rabies ELISA sera	Positive (≥70%)	62	17			37	9		
	Negative (<70%)	0	54	87.2%	(80.3–92.4)	0	37	89.2%	(80.0–94.9)
	Total	62	71			37	46		

**Table 3 viruses-14-00711-t003:** Overall agreement between FP eluates titrated with the BioPro ELISA kit versus sera titrated with FAVN test or with ELISA test.

Assay	Result	BioPro Rabies ELISA Ab Kit—FP Eluates			
		Dogs—Experimental and Field Conditions (France and Tunisia)			
		Positive (≥70%)	Negative (≥70%)	Overall Agreement	95% CI
FAVN test sera	Positive (≥0.50 IU/mL)	93	18		
	Negative (<0.50 IU/mL)	6	99	88.9%	(83.9–92.8)
	Total	99	117		
BioPro Rabies ELISA sera	Positive (≥70%)	99	26		
	Negative (<70%)	0	91	88.0%	(82.9–92.0)
	Total	99	117		

## Data Availability

The data presented in this study are available on request from the corresponding author.

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
