# Peer review of "Filter Papers to Collect Blood Samples from Dogs: An Easier Way to Monitor the Mass Vaccination Campaigns against Rabies?"

_viruses, 2022, doi:10.3390/v14040711_

Round 1

Reviewer 1 Report

In this manuscript the authors describe collection of blood on filter paper for subsequent use in serological assays to monitor rabies vaccine responses in dogs. Overall, the manuscript is well written. My major concern is regarding the methodology/discussion. Throughout the manuscript the authors refer to new methodology, but using filter paper, ELISA or FAVN is not new. However, in the discussion section the authors refer to the new collection method: i.e. “consisting to apply a piece of filter paper on the wound of the animal to collect blood without needle or syringe” (lines 279-280). The authors refer to this collection method several times in the discussion, and this is confusing since this “method” is not described anywhere else in the manuscript. In the blood sample collection section, lines 103-121, the authors specifically mention that blood was taken from the jugular or cephalic vein into two untreated tubes – presumably with a needle or syringe (this information is not specified). Lines 354-355 also state that inflicting wounds on a pet dog would be unacceptable to owners – so how would this collection method then be implemented? The discussion section does not seem to follow what is described in the method section - this is very confusing and as a result after reading the entire manuscript the reader is unclear as to what it is actually about - the FAVN, ELISA, FP or collection method?

Minor comments:

Line 19: The authors refer to a “new tool” – this is unclear as to what they are referring to, the collection method?

Lines 107, 116, 353: For the study the authors state that 10 x 7.5 cm FP was saturated with whole blood, approximately 7.5 ml blood. In the discussion section the authors mention that only 1-1.5 ml of blood will be needed. The authors should please include data to support this statement – what are the minimum and maximum volumes that will be needed to provided consistent results. Is there more variation in the results when different volumes are used, for example 1 ml vs 5 ml?

Lines 344-345: The authors state “This is not less painful for the dog…”. This states that collecting blood from a wound after making a cut in the ear is more painful for the dog – how is this then an improvement?

Author Response

Thank you for reviewing this article.

Below are our replies for carefully responding to all the comments. The paper has also been corrected accordingly.

Regarding the "Comments and Suggestions for Authors
":

In this manuscript the authors describe collection of blood on filter paper for subsequent use in serological assays to monitor rabies vaccine responses in dogs. Overall, the manuscript is well written. My major concern is regarding the methodology/discussion. Throughout the manuscript the authors refer to new methodology, but using filter paper, ELISA or FAVN is not new. However, in the discussion section the authors refer to the new collection method: i.e. “consisting to apply a piece of filter paper on the wound of the animal to collect blood without needle or syringe” (lines 279-280). The authors refer to this collection method several times in the discussion, and this is confusing since this “method” is not described anywhere else in the manuscript. In the blood sample collection section, lines 103-121, the authors specifically mention that blood was taken from the jugular or cephalic vein into two untreated tubes – presumably with a needle or syringe (this information is not specified). Lines 354-355 also state that inflicting wounds on a pet dog would be unacceptable to owners – so how would this collection method then be implemented? The discussion section does not seem to follow what is described in the method section - this is very confusing and as a result after reading the entire manuscript the reader is unclear as to what it is actually about - the FAVN, ELISA, FP or collection method?

Response : This method, consisting of testing blood samples deposited on Filter papers, is used for the first time in dogs, that is why we have spoken about a “new method”. The method includes the use of Filter paper (FP) for collecting blood and the ELISA test for the rabies serological testing. The tool is the FP for collecting the blood combined with the use of a serologic test on blood to assess the humoral immune response of dogs. Another study will assess the faisability of applying the FP directly on a small wound.   

The discussion has been revised to take into consideration the confusion mentioned regarding the method (see Lines 353 – 365). As regards other points underlined, as they are also reported in the paragraph “Minor comments”, please see below our answers and proposed changes.

Minor comments :

1/ Line 19: The authors refer to a “new tool” – this is unclear as to what they are referring to, the collection method?

Response : the Line 19 states that the study explores the performance of “a new tool using FP to collect blood samples associated with an ELISA”. The association FP – ELISA is indeed new for rabies antibody testing of dogs. The collection method (which is new also for dogs rabies serology) is closely associated with the ELISA (which is not new), as the protocol of treatment of the FPs is developed by using the eluates tested by ELISA (the cell layers being altered when tested by a seroneutralisation test on cells - see Lines 142 – 143).

2/ Lines 107, 116, 353: For the study the authors state that 10 x 7.5 cm FP was saturated with whole blood, approximately 7.5 ml blood. In the discussion section the authors mention that only 1-1.5 ml of blood will be needed. The authors should please include data to support this statement – what are the minimum and maximum volumes that will be needed to provided consistent results. Is there more variation in the results when different volumes are used, for example 1 ml vs 5 ml?

Response : For this preliminary study, we used a 10 x 7.5 cm FP saturated with a minimal volume of 7.5mL to be able to perform other tests/controls if necessary. This minimum volume is sufficient to saturate entirely the whole FP as tested for wildlife by Wasniewski et al, 2014. For analyses in the field, one strip of FP (1 x 7.5 cm) is sufficient to perform the rabies serological control. To impregnate this small strip of FP we need a lower volume of blood than for the whole FP. That is why we wrote between 1 and 1.5 mL to be sure that the strip of FP will be easily impregnated in the field. Once the FP is totally soaked, there is no variation of the results according to the volume used to impregnate the strip of FP. When the FP is completely soaked even if you add more blood, you cannot it impregnate more and the results are not impacted. The key point when using FP for collecting blood is that the strip shall be completely soaked before drying to be able to obtain consistent results.

To respond to the remark, in Lines 109 and 115, we have inserted precisions on the method for collecting the blood. We did not use is this study the application of the FP paper directly on the wound as the objective was to assess the feasibility of testing FP blood papers with an ELISA.

3/ Lines 344-345: The authors state “This is not less painful for the dog…”. This states that collecting blood from a wound after making a cut in the ear is more painful for the dog – how is this then an improvement?

Response : This method of blood sampling associated with an ELISA would be particularly adapted for uncontrollable and free roaming dogs. We have improved the discussion (see Lines 364 – 365).

Reviewer 2 Report

Line 63: "…blood vials "in" a laboratory…" should be …"to" a laboratory….

Line 176: as the abbreviation was mentioned before "FAVN test" could be shortened "FAVNt".

Line 141-142: There is probably a gap/paragraphing error. The sentence starting with "this step…" gives the impression that there has been another sentence before. Otherwise, it doesn't make sense.

Table 1: Why does total number of results given for animals from France correspond to the total number of animals (133) but total number of results given for animals from Tunisia correspond to the total number of samples: 83. (Number of animals from Tunisia: 29). Data presentation is inconsistent here.

Table 2: Might total numbers in the row of the BioPro Rabies ELISA ab-kit FP eluates be wrong? Because for the positive results: should be 62+0=62 but it's written 61. Same for the negatives: 17+54=71 but it's given 21.

Line 188 & 297-298: That one eluate is against most important outcome of the study: high specificity! As the serum of the associated eluate was found to be negative by both tests, the result of this eluate is a false positive and it's not discussed anywhere why this occurred.

Line 271: "…handling of blood samples on non-obedient dogs." should be corrected as " handling of non-obedient dogs during blood sampling".

Line 316-319: The statement that experimental conditions are better for a higher percentage of agreement between tests wouldn't be valid for the comparison of ELISA results from France and Tunisia, as the ELISA's performed on the samples from Tunisia revealed higher percentage of agreement.

Line 337: "no false positive" is a wrong statement for this study, as there has been 1 false positive eluate.

Line 343: Is sampling by cutting the ear was done in another study (probably yes)? It would be easier to understand if stated accordingly.

Line 359: It's not clear what the recommended method for blood collection is. If the final recommended method is to collect the blood into the tubes and impregnate the FPs in the tube, in deed special equipment is needed (syringe+serum tubes). If the recommended method is to make a puncture on the cephalic/jugular vein and impregnate the FP with the blood leaking from the puncture, special equipment is still need (blade to shave the hair+lancet). Either method is not easier than taking blood for normal serology. So the statement in the previous parts of the text claiming sampling via FP is easier and safer in comparison to regular blood sampling is not quite right. The only advantage of the FPS would be the ease in storage and transportation.

General remarks to the paper:

  • How were the results based on different sampling days from the same animal? Is there a higher agreement between test results after distinct day post-vaccine? If it's not going to contribute to the outcome of the study, what's the point in taking several samples on different days post vaccination?
  • Which vaccines were used for the dogs in France and Tunisia? Are they the same product? This information might contribute to interpretation of the differences in agreement percentages of the tests.
  • Are FPs sterile? This is important to know/to write in the paper to exclude bacterial degradation of antibodies, which otherwise probably might explain the negative eluates that were found to be positive in other tests.

The reasons for some results are not evaluated in depth:

  • What's the explanation for differences in results from different geographical origin?
  • Is the fact considered, while interpreting FAVN negative but ELISA positive samples, that FAVN test measures neutralizing antibodies whereas ELISA not only neutralizing but also other additional antibodies?

Author Response

Thank you for reviewing this article.

Below are our replies for carefully responding to all the comments. The paper has also been corrected accordingly.

1 / Line 63: "…blood vials "in" a laboratory…" should be …"to" a laboratory….

Response : The correction has been done in the manuscript.

2 / Line 176: as the abbreviation was mentioned before "FAVN test" could be shortened "FAVNt".

Response : the change has been done for all the manuscript.

3 / Line 141-142: There is probably a gap/paragraphing error. The sentence starting with "this step…" gives the impression that there has been another sentence before. Otherwise, it doesn't make sense.

Response: the word « step » has been replaced by “processing” in the text.

4 / Table 1: Why does total number of results given for animals from France correspond to the total number of animals (133) but total number of results given for animals from Tunisia correspond to the total number of samples: 83. (Number of animals from Tunisia: 29). Data presentation is inconsistent here.

Response : There is a misunderstanding, 133 corresponds to the number of samples and not to the number of animals from France. Indeed as explained in lines 79 to 83: 111 dogs were collected in the Anses-Nancy experimental station and according to some experimental protocols, 22 vaccinated dogs were sampled twice, resulting in 133 samples. So, in this table 1, the data concern the number of samples both in France and in Tunisia.

5 / Table 2: Might total numbers in the row of the BioPro Rabies ELISA ab-kit FP eluates be wrong? Because for the positive results: should be 62+0=62 but it's written 61. Same for the negatives: 17+54=71 but it's given 21

Response : We made a mistake, the values of total numbers were corrected in Table 2.

6 / Line 188 & 297-298: That one eluate is against most important outcome of the study: high specificity! As the serum of the associated eluate was found to be negative by both tests, the result of this eluate is a false positive and it's not discussed anywhere why this occurred.

Response : For this sample, there is a misunderstanding as the eluate as well as the corresponding serum gave a positive result by using the ELISA as described in the Result part. However we made a mistake in the discussion by writing that the corresponding serum sample was found negative by both methods. We have corrected the sentence (see Line 339). A specificity of 98% is considered for rabies serology as high, particularly with field samples collected for assessing herd immunity (see Lines 303 – 305).

7 / Line 271: "…handling of blood samples on non-obedient dogs." should be corrected as " handling of non-obedient dogs during blood sampling".

Response : This has been corrected in the text.

8 / Line 316-319: The statement that experimental conditions are better for a higher percentage of agreement between tests wouldn't be valid for the comparison of ELISA results from France and Tunisia, as the ELISA's performed on the samples from Tunisia revealed higher percentage of agreement.

Response : The reviewer is right. By writing that, we have only focused our discussion on the results obtained between the FP eluates and the corresponding serum tested by FAVNt. We have modified our sentence to make it clearer. Most of time, the samples obtained from experimental conditions have a better “quality” than those collected in the field. As seroneutralisation assay use live cells and rabies virus, the FAVNt is much more sensitive to the “quality” of the samples than the ELISA.

9 / Line 337: "no false positive" is a wrong statement for this study, as there has been 1 false positive eluate.

Response : The sentence has been modified to include the false positive result obtained for the eluate comparing to the FAVNt result on the corresponding serum.

10 / Line 343: Is sampling by cutting the ear was done in another study (probably yes)? It would be easier to understand if stated accordingly.

Response : The method of blood sampling at the ear (with a needle) is currently used in rodents for laboratory purposes. We did not find studies reporting ear cutting for further serological work in field dogs, and the feasability will be assessed. However, even if difficult or impossible, collecting blood with a syringe and a needle and impregnating a strip of filter paper presents advantages (see below).

11 / Line 359: It's not clear what the recommended method for blood collection is. If the final recommended method is to collect the blood into the tubes and impregnate the FPs in the tube, in deed special equipment is needed (syringe+serum tubes). If the recommended method is to make a puncture on the cephalic/jugular vein and impregnate the FP with the blood leaking from the puncture, special equipment is still need (blade to shave the hair+lancet). Either method is not easier than taking blood for normal serology. So the statement in the previous parts of the text claiming sampling via FP is easier and safer in comparison to regular blood sampling is not quite right. The only advantage of the FPS would be the ease in storage and transportation.

Response : We agree with this comment and we have modified the sentence accordingly (see Lines 369 – 370).

General remarks to the paper :

1 / How were the results based on different sampling days from the same animal? Is there a higher agreement between test results after distinct day post-vaccine? If it's not going to contribute to the outcome of the study, what's the point in taking several samples on different days post vaccination?

Response : The results based on different sampling days from the same animal is not going to contribute to the outcome of the study, that is why these results are neither used nor specifically presented in this paper.

2 / Which vaccines were used for the dogs in France and Tunisia? Are they the same product? This information might contribute to interpretation of the differences in agreement percentages of the tests.

Response : As the main aim of this preliminary was to evaluate the ability of this method to detect rabies antibodies elicited after a parenteral vaccination, the choice of the rabies vaccines was not a key parameter. Moreover and more importantly, the status of the sample (negative or positive) was determined on the serum by reference to the FAVNt. Two different licensed vaccines were used in the experimental protocols performed in France which were different from the one used during the vaccination campaign in Tunisia. From an ethical point of view, to reduce the number of animals involved in experimental studies, the animals used in this study, belonged to other protocols and we took the opportunity of these protocols to collect blood samples in parallel for own.

3 / Are FPs sterile? This is important to know/to write in the paper to exclude bacterial degradation of antibodies, which otherwise probably might explain the negative eluates that were found to be positive in other tests.

Response : The FPs were not sterile as they were simply air-dried after soaking. It was not possible to perform this work on the field under sterility conditions. The aim of using FP is to simplify the method of blood collection as it was already done for wildlife including bats. By using the ELISA test combined to the FP blood collection, even if the sample are of “poor” quality, the results remained reliable. Indeed, ELISA are well-known to better suit to poor-quality, haemolysed, relatively cytotoxic and potentially contaminated wildlife specimens (Cliquet F, Sagne L, Schereffer JL, Aubert MF. ELISA test for rabies antibody titration in orally vaccinated foxes sampled in the fields. Vaccine. 2000;18:3272–9.). However, we cannot exclude the fact that a possible bacterial contamination has occurred on the FP degrading the level of rabies antibodies and resulting in false negative results. However if we compared the results of the serum samples only (without any potential bacterial contamination) testing by both methods (supplementary file S1), we also have an agreement around 87%.

4 / What's the explanation for differences in results from different geographical origin?

Response : Regarding the differences in results from different geographical, it could be explained by samples collected experimentally and samples collected in the field. We mentioned this hypothesis in our discussion.

5 / Is the fact considered, while interpreting FAVN negative but ELISA positive samples, that FAVN test measures neutralizing antibodies whereas ELISA not only neutralizing but also other additional antibodies?

Response : a sentence has been inserted to point this important comment (Lines 338 – 340).  

Round 2

Reviewer 1 Report

Authors have addressed previous comments

Author Response

Thank you.